# Genome-Based Exploration of *Rhodococcus* Species for Plastic-Degrading Genetic Determinants Using Bioinformatic Analysis

**DOI:** 10.3390/microorganisms10091846

**Published:** 2022-09-15

**Authors:** Jessica Zampolli, Alessandro Orro, Daniele Vezzini, Patrizia Di Gennaro

**Affiliations:** 1Department of Biotechnology and Biosciences, University of Milano-Bicocca, Piazza della Scienza 2, 20126 Milan, Italy; 2Institute of Biomedical Technologies, National Research Council, CNR, Via Fratelli Cervi 19, 20133 Segrate, Italy

**Keywords:** plastic, polymer biodegradation, *Rhodococcus* genus, genome analyses, plastic-degrading enzymes, oxidase, esterase, PET-hydrolase, depolymerase, hydroxylase/monooxygenase

## Abstract

Plastic polymer waste management is an increasingly prevalent issue. In this paper, *Rhodococcus* genomes were explored to predict new plastic-degrading enzymes based on recently discovered biodegrading enzymes for diverse plastic polymers. Bioinformatics prediction analyses were conducted using 124 gene products deriving from diverse microorganisms retrieved from databases, literature data, *omic*-approaches, and functional analyses. The whole results showed the plastic-degrading potential of *Rhodococcus* genus. Among the species with high plastic-degrading potential, *R*. *erythropolis*, *R*. *equi*, *R*. *opacus*, *R*. *qingshengii, R*. *fascians*, and *R*. *rhodochrous* appeared to be the most promising for possible plastic removal. A high number of genetic determinants related to polyester biodegradation were obtained from different *Rhodococcus* species. However, score calculation demonstrated that *Rhodococcus* species (especially *R*. *pyridinivorans*, *R*. *qingshengii*, and *R*. *hoagii*) likely possess PE-degrading enzymes. The results identified diverse oxidative systems, including multicopper oxidases, alkane monooxygenases, cytochrome P450 hydroxylases, *para*-nitrobenzylesterase, and carboxylesterase, and they could be promising reference sequences for the biodegradation of plastics with C−C backbone, plastics with heteroatoms in the main chain, and polyesters, respectively. Notably, the results of this study could be further exploited for biotechnological applications in biodegradative processes using diverse *Rhodococcus* strains and through catalytic reactions.

## 1. Introduction

Within Actinobacteria phylum (Corynebacteriaceae order), *Rhodococcus* represents one of the most extraordinary genera in terms of catabolic versatility, since the members of this genus deploy an extensive asset of genes encoding enzymes enabling the biotransformation and/or biodegradation of a wide array of organic compounds, contaminants, pharmaceuticals, and wastes [1,2,3,4,5,6,7,8,9]. The remarkable metabolic capabilities of *Rhodococcus* species together with their strong persistence under difficult conditions are due to their peculiar cell features including the cell envelope (containing mycolic acids, and peptidoglycan), hydrophobicity, a specific phospholipid profile, and large and plastic genomes with a multiplicity of genes for catabolic and anabolic processes, and stress response [10,11,12,13,14]. In line with this aspect, rhodococci can inhabit different environments from groundwater to water systems, soils, diverse extreme and harsh environmental niches, and contaminated ecosystems; they have been also isolated from plants (*R*. *fascians*), insects (*R*. *rhodnii*), diseased and healthy animals (e.g., *R*. *equi*), and humans (e.g., *R*. *equi*, *R*. *rhodochrous*, and *R*. *erythropolis*) [15,16]. However, until now, *Rhodococcus* has received relatively little attention for plastic biodegrading capabilities of diverse biodegradable and non-biodegradable polymers and the corresponding genetic determinants [17,18,19].

Synthetic plastic constitutes a large group of compounds with profitable features, e.g., strength, flexibility, extreme durability, low weight, and easy and low-cost production, that are crucial in everyday life, which led to a global plastic production of almost 360 million tonnes in 2018 that is still growing year by year [20]. Synthetic plastics are commonly grouped into two main categories, based on their chemical structure: (i) the most recalcitrant carbon−carbon (C−C) backbone plastics, including low and high molecular density polyethylene (PE), and polypropylene (PP); (ii) the most prone to biodegradation, heteroatomic backbone plastics, including polyethylene terephthalate (PET) and polyurethane (PU) [21,22].

As a consequence of plastic utilization, emerging issues in disposal management are rising and are becoming an extreme priority. Up to now, environmental plastic pollution has been considered merely a facade disturbance, but recent research revealed potential risks of plastic impacts on biota in diverse ecosystems and on human health [23]. The actual plastic waste crisis, predicting around 12 thousand million metric tons of plastic waste accumulating in landfills and the natural environment by 2050 [24], seeks new technologies for recycling post-consumer plastics for waste valorization and/or the optimization of the biodegradation of discarded plastic polymers. Biocatalytic degradation and/or biotransformation by enzymes is an efficient and sustainable alternative for synthetic plastic (usually considered non-biodegradable) depolymerization to support and complement plastic recycling with mechanical, chemical, or chemical-physical treatments [25]. 

On the other hand, all polyester-based plastics are considered theoretically biodegradable, since the esterification process for their production is chemically reversible, even by biocatalysis [26]. This category includes, among others, (i) aliphatic polyesters, such as polyhydroxyalkanoates (PHAs), poly(propiolactone) (PPL), poly(ε-caprolactone) (PCL), poly(L-lactic acid) (PLA), and poly(ethylene succinate) (PES); (ii) co-polyesters containing aliphatic and aromatic components, such as poly(butylene succinate) (PBS), poly(butylene succinate)-co-(butylene adipate) (PBSA), poly(butylene adipate-co-terephthalate) (PBAT), poly(butylene succinate- co-terephthalate) (PBST), and poly(butylene succinate/terephthalate/isophthalate)-co-(lactate) (PBSTIL) [27]. 

Diverse plastic-degrading enzymes have been discovered from microbial sources, but in-depth investigations are still necessary for biotechnological industrial applications [28]. Despite the lack of enzymes able to oxidize highly stable C−C bonds of polyolefins hampering their biodegradation, few of them were retrieved from transcriptomic studies and/or functional analysis, including hydroxylases/monooxygenases, oxidases, laccases, and peroxidases [17,29,30,31]. Among the few plastic-degrading enzymes closest to achieving feasible biotechnological industrial processes, PET hydrolases are the most prominent/valuable, although it is still unknown how frequently they appear in different bacterial species. The majority of this enzyme category comprises cutinases, showing a broad substrate specificity, lipases, possessing a lower activity that is generally specific for long-chain substrates, and esterases, which usually hydrolyze esters with a shorter aliphatic chain in comparison to lipases [32].

Different PU-degrading enzymes were also reported (e.g., cutinases, esterases, lipases, laccases, peroxidases, proteases, and ureases) [25,33,34], but most of them showed ester-linked PU activity and were unlikely to show ether-linked PU activity [35]. 

In general, a single specific enzyme is not able to catalyze the biodegradation of a certain plastic group, but enzyme categories are recognized to be able to perform a step of polymer biotransformation, and the overall biodegradation can occur only due to the involvement of a multiplicity of enzymatic arrays and their subsequent activity [21,28].

Since *Rhodococcus* members play an important role in biotechnological applications [7], and only a few rhodococci have been revealed to possess plastic-degrading abilities and in a few cases their genetic determinants have been investigated [17,18,19], the present study investigates for the first time 669 *Rhodococcus* genomes mining predictive plastic-degrading enzymes. Comparative bioinformatics analyses were performed using 124 gene products related to plastic degradation of diverse polymers from different bacterial genera; they were retrieved from databases, literature data, *omic*-approaches, and functional analyses. The level of similarity of these enzymes from other organisms against *Rhodococcus* genomes in connection with the analysis of clustering contributes to evaluate the evolutional relationships and/or the putative horizontal gene transfer of new *Rhodococcus* identified sequences. In addition, score calculation provides a weight level to evaluate the real potential role of the different *Rhodococcus* species and different genes involved in the metabolism of diverse polymers.

## 2. Material and Methods

### 2.1. Input Data 

Predictive analyses were performed on 669 *Rhodococcus* genomes from NCBI (http://www.ncbi.nlm.nih.gov, accessed on 27 May 2022) and analyzed by blast search.

*Rhodococcus* genomes were screened for 124 gene products related to plastic degradation of diverse polymers from different microorganisms retrieved from PlasticDB database [36], literature data, *omic* studies, and functional analyses in order to investigate the plastic-degrading potential of *Rhodococcus* genus (Appendix A). The considered gene products were reported to be involved in the biodegradation and/or biotransformation of at least one considered plastic polymer. Among the diverse polymers, synthetic (non-biodegradable) or biodegradable plastics taken into account were organized into three major groups: plastics with C−C backbone comprising PE, plastics with heteroatoms in the main chain, including PET and PU, and polyesters (both aliphatic and co-polyesters containing aliphatic and aromatic components), such as polyhydroxybutyrate (PHB), PES, PPL, PCL, PLA, (poly(3-hydroxyvalerate) (PHV), poly(3-hydroxybutyrate-co-3-hydroxyvalerate) (PHBV), poly(hydroxyphenyl-valerate) (PHPV), PBS, PBSA, PBAT, and Ecoflex. 

The unique input gene product sequences were also aligned with Clustal Omega program using the multiple sequence alignment (MSA) tool with default parameters (Neighbor-Joining method, the Gonnet transition matrix, gap opening penalty of 6 bits, maintain gaps with an extension of 1 bit, used bed-like clustering during subsequent iterations, and zero number of combined iterations) [37] in order to establish a clusterization framework of gene products used as input against *Rhodococcus* genomes. MSA was used as input of cluster analysis inferred using the maximum likelihood (ML) method by MEGA (version 10.2) software [38], with the following settings: JTT substitution matrix and gamma distribution of mutation rates with gamma optimized to 2. As a test of inferred phylogeny, 50 bootstrap replicates were used.

### 2.2. Comparative Analyses for Plastic-Degrading Gene Products in Rhodococcus Species

A list of 124 gene products known to be involved in the degradation of plastic polymers from different microorganisms was used as a reference for blast search (https://blast.ncbi.nlm.nih.gov/Blast.cgi, accessed on 27 May 2022) against the *Rhodococcus* dataset (Taxid 1827) of genomes. The outcomes of the analyses were defined as HIT sequences, i.e., all the output sequences presenting a degree of similarity with respect to the query sequence. The threshold is 5% and it is expressed as an e-value: if the statistical significance of the match (query against the random model) is greater than 5%, the match is not reported. The hit sequences of all results have been collected and annotated according to the following features in a single table: Gene ID, Plastic Group, Plastic, Plastic Enzyme and *Rhodococcus* Species, and Blast output (Score, *e*-value, and Identity).

Statistical analysis has been performed in order to count the number of normalized unique genes identified in *Rhodococcus* genomes grouped by different criteria (by plastic type, *Rhodococcus* species, enzymes, and others), as reported in the resulting plots. In addition, the score calculation has been obtained as a sum of Blast Score over all the groups of selected genes identified in *Rhodococcus* genomes and it indicates the plastic degradation potential of the diverse *Rhodococcus* species; manual curation was performed using the BLASTx tool of NCBI pipeline [39], Clustal Omega [37], Protein Data Bank [40], UniProt [41], and KEGG [42].

### 2.3. Rhodococcus HIT Sequence Tree Clusterization Analysis

On the basis of the bioinformatics analyses for the search for plastic-degrading gene products, the identified *Rhodococcus* genetic determinants were aligned by BLASTp tool [39] to determine sequence homology and retrieve the corresponding sequences. Reference proteins were selected among 124 gene products used as input data selected for the plastic-degrading ability from different microorganisms. 

The sequences identified in *Rhodococcus* genomes belonging to multicopper oxidases, alkane 1-monooxygenases, and cytochrome P450 hydroxylases putatively involved in C−C backbone plastics (PE) degradation, and gene products putatively involved in plastics with heteroatoms in the main chain (PET and PU) degradation were aligned on Clustal Omega program using the MSA tool with default parameters [37], using the corresponding reference sequences from other microorganisms. For multiple sequence alignment with more than 4000 sequences, MAFFT software v. 7 with default parameters has been used [43]. NCBI Tree Viewer 1.19.2 was used to generate the corresponding tree clusterization images.

In order to generate subtree clusterization, MSA was used as input of cluster analysis inferred using the maximum likelihood (ML) method by MEGA (v. 10.2) software [38], with the same setting used for the unique input gene product sequences clusterization. The resulting groups allowed to define different trees, showing clades with putative functions identified by the UniProt database [41].

## 3. Results

### 3.1. Overview of Rhodococcus Genus Potential for Plastic Degradation

In order to explore the genetic determinants related to plastic degradation of *Rhodococcus* genomes, diverse gene products deriving from other microorganisms were retrieved from literature data, genomic and transcriptomic studies, and the PlasticDB database [36]. In total, 124 input gene products (Appendix A) were listed for specific polymer degradation capabilities, including the degradation of plastics with C−C backbones, plastics with heteroatoms in the main chain, and polyesters. Among them, 75 sequences were unique and belong to Gram-negative bacteria, Gram-positive bacteria, and fungal strains. Since only a few studies investigate the genetic determinants of *Rhodococcus* strains for plastic degradation, other microorganisms were considered to investigate in-depth the plastic biodegrading potential of *Rhodococcus* genus. Within these microorganisms, 27 bacterial genera and 5 fungal genera belong, respectively, to four different bacterial phyla (Actinobacteria, Bacteroidetes, Firmicutes, and Proteobacteria) and two fungal phyla (Aspergillus and Cryptococcus). In order to establish a clusterization framework of gene products used as input against *Rhodococcus* genomes, 75 unique input gene product sequences were considered. The generated tree reveals that these amino acid sequences for polymer degradation capabilities are divided into fifteen clades (Figure 1): cutinase I (clade I), polyester-hydrolase/PET-hydrolase (clade II), depolymerase (clade III), oxygenase (clade IV), multicopper oxidase (clade V), cytochrome P450 hydroxylase (clade VI), PLA-depolymerase (clade VII), esterase (clade VIII), PU esterase (clade IX), membrane transporters (clade X), carboxylesterase (clade XI), PHB-depolymerase I (clade XII), polyurethanase (clade XIII), cutinase II (clade XIV), and PHB-depolymerase II (clade XV).

These considered genes deriving from different microorganisms were reported for their involvement in the degradation of the following polymers, classified into three main categories: (i) C−C backbone plastics, including PE; (ii) heteroatomic backbone plastics including PET and PU; (iii) polyesters including PHB, PES, PPL, PCL, PLA, and co-polyesters containing aliphatic and aromatic components such as PHV, PHBV, PHPV, PBS, PBSA, PBAT, and Ecoflex.

For every polymer type, at least one plastic-degrading gene (retrieved from the PlasticDB database, literature data, *omic* studies, and functional analyses) was used for the comparative analysis against *Rhodococcus* genomes, and, in case of more than one gene from different microorganisms described for their ability to biodegrade a specific plastic polymer, multiple genes were selected to broaden the spectrum of phyla used in the overall comparison.

After aligning all 124-input degrading-genes against 669 *Rhodococcus* genomes, the total number of identified sequences was almost 50,000 (Appendix A), and within them, only the unique sequences were further considered. They were classified into three main polymer categories as shown in Figure 2. The highest number of sequences was in the polyester group (16,528 sequences) followed by C−C backbone plastics and heteroatomic backbone plastics groups (respectively 7069 and 5340 sequences, respectively). This is in line with the highest number of input-degrading genes related to the polyester group and also their general higher biodegradability.

Considering each polymer, the highest number of *Rhodococcus* DNA sequences encodes proteins putatively related to the degradation of PE, PBAT, PHB, PET, and PU; a few genes cover PHBV, PHV, PHPV, and PPL degradation (Figure 3). 

Overall, the results show the biodegradative potential of the *Rhodococcus* genus towards both synthetic and biodegradable plastics. Furthermore, these results evidenced a relation between gene products and the different species of *Rhodococcus* genus (Appendix A). 

Among *Rhodococcus* species, the highest number of genes putatively involved in plastic degradation belongs to *R*. *equi*, *R*. *erythropolis*, *R*. *opacus*, *R*. *qingshengii*, *R*. *fascians*, and *R*. *rhodochrous* (with 2643, 2464, 1786, 1777, 1640, and 1215 predicted genetic determinants, respectively). The lowest numbers of sequences predicted for plastic degradation are related to the following species: *R*. *baikonurensis*, *R*. *artemisiae*, *R*. *nanhaiensis*, *R*. *sovatensis*, *R*. *canchipurensis*, *R*. *jialingiae*, and *R*. *percolatus* (Figure 4). Nevertheless, these data should be evaluated considering the different number of genomes of species of the *Rhodococcus* genus.

Moreover, these species present gene products mainly predicted for C−C backbone plastic degradation (Figure 5). Likewise, the category “uncultured *Rhodococcus* sp.” showed a few numbers of sequences predicted for plastic degradation and primarily against C-C backbone plastics. Among all the species, the average trend of unique HIT sequence number shows more than 50% of polyester degrading gene products, except for the species possessing the lowest number. The remaining 50% is mostly covered by PE-degrading gene products (Figure 5). 

### 3.2. Rhodococcus Genus Degradative Potential towards C-C Backbone Polymer 

The degradative potential of the *Rhodococcus* genus towards C-C backbone polymers was evaluated including only PE-degrading genes, since other polymers, such as polypropylene (PP), were correlated to similar genes associated with PE-degradation [44], or polystyrene (PS) and polyvinyl chloride (PVC), whose biodegradation is supported by few studies that did not focus on the molecular aspects and genetic determinants for these metabolisms [21]. The considered input genes encode alkane 1-monooxygenases, multicopper oxidases, cytochrome P450 hydroxylases, and transporters (possible membrane protein, integral membrane protein, and possible conserved integral membrane protein). The results of the bioinformatic analysis are shown in Figure 6 where the number of unique HIT sequences identified in *Rhodococcus* genomes is reported for different enzyme categories putatively related to C−C backbone plastic degradation. Figure 6 shows that the highest number of unique HIT sequences identified in *Rhodococcus* species belongs to the cytochrome P450 hydroxylase group (4864 sequences).

The clusterization analysis of all identified HIT sequences was performed by comparing multicopper oxidases, alkane monooxygenases, or cytochrome P450 hydroxylases, which were the most abundant sequences for their potential degradative role. They were clusterized into three different trees including only the unique HIT sequences of *Rhodococcus* that were 816, 940, and 1719 for the multicopper oxidase, alkane monooxygenase, and cytochrome P450 hydroxylase clusterization tree, respectively (Appendix A). For each tree, the HIT sequences with the highest similarity with respect to the reference sequences were considered for the subtree generation. The considered reference sequences for the multicopper oxidase category were AII08809, AII11185, and AII11221, for alkane monooxygenase category it was AII08632, and for cytochrome P450 hydroxylase category it was AII08421. The first subtree showed mostly copper and multicopper oxidases (MmcO) belonging to the following species: *Rhodococcus* sp., *R*. *opacus*, *R*. *wratislaviensis, R*. *marinonascens*, *R*. *imtechensis*, *R*. *jostii*, and *R*. *koreensis* (Figure 7). The reference sequences clusterized into three different clades: AII08809 (multicopper oxidase I), grouped with a multicopper oxidase of *R*. *wratislaviensis* (WP_037231000) and a FtsP/CotA-like multicopper oxidase with cupredoxin domain of *R*. *wratislaviensis* (WP_112302511); AII11185 (multicopper oxidase II), clusterized with the clade of AII11221 reference; and AII11221 (multicopper oxidase III), grouped with two multicopper oxidases of *Rhodococcus* sp. (WP_095862380 and WP_133987587). 

Considering the alkane monooxygenase subtree, eight clades can be distinguished comprising only alkane 1-monooxygenase (AII08632), belonging to the same species observed in the previous subtree (Figure 7) with the addition of *R*. *rhodochrous* and *R*. *aetherivorans* species (Figure 8). The closest sequences with respect to the reference input AII08632 belong to the *R*. *wratislaviensis* and *R*. *jostii* species.

Cytochrome P450 hydroxylase subtree showed sequences belonging to *Rhodococcus* sp., *R*. *opacus*, *R*. *wratislaviensis*, *R*. *imtechensis*, *R*. *jostii*, and *R*. *koreensis*, which clusterized into eight clades (Figure 9). The reference sequence AII08421 principally clusterized with cytochrome P450 of *R*. *wratislaviensis* (WP_112300243) and the closest clade include cytochrome P450 of all the species mentioned in the previous subtrees except *R*. *jostii* (WP_012691738, WP_009475556, WP_144285016, WP_124393366, WP_095862123, WP_072937090, WP_185950062, WP_007299636, WP_087560732, WP_218267768, and WP_206009113).

### 3.3. Degradative Potential of the Rhodococcus Genus towards the Degradation of Plastics with Heteroatoms in the Main Chain 

*Rhodococcus* degrading potential for plastics with heteroatoms in the main chain was evaluated by selecting PET and PU degrading genes as references. The considered input genes encode cutinase, PU esterase, PET-hydrolase, triacylglycerol-lipase, polyester-hydrolase, polyurethanase, serine-hydrolase, and *para*-nitrobenzylesterase. The results of the comparison of *Rhodococcus* genomes against these reference sequences are shown in Figure 10, where the number of unique HIT sequences identified in *Rhodococcus* genomes is reported for different enzyme categories putatively related to heteroatomic backbone plastic degradation. The highest number of unique HIT sequences identified in *Rhodococcus* species belongs to PU esterase and *para*-nitrobenzylesterase groups (3205 and 2561 unique sequences, respectively).

In order to evaluate the potential of the diverse enzymes for PET and PU degradation, clusterization analyses of respectively 2938 and 2118 unique HIT sequences identified in *Rhodococcus* genomes for the two different polymers revealed two clusterization trees against the reference gene inputs (Appendix A). Considering PET-degrading enzymes used as input sequences, a subtree clusterization of 110 HIT sequences was performed (Appendix A). Gene products classified as cutinase, PET-hydrolase, triacylglycerol-lipase, polyester-hydrolase, and serine-hydrolase putatively involved in PET degradation were included. *para*-Nitrobenzylesterases were excluded for the high number of HIT sequences and because they were clusterized on their own.

For PU-degrading genes, a PU subtree was generated. The most similar HIT sequences with respect to the reference sequences were considered for subtree generation (Figure 11). The figure shows the main clusters obtained. Sequences clustering closer to BAA76305 (PU esterase) reference protein were mostly deriving from *Rhodococcus* sp., *R*. *opacus*, *R*. *wratislaviensis*, *R*. *jostii*, *R*. *aetherivorans,*
*R*. *hoagii, R. erythropolis*, and *R*. *qingshengii*. The reference sequence WP_088276085 (Polyester hydrolase) principally clusterized with *Rhodococcus* sp., *R*. *rhodochrous*, *R*. *zopfii*, *R*. *triatomae*, and *R*. *fascians*.

### 3.4. Rhodococcus Genus Degradative Potential towards Polyesters

Polyesters are often considered biodegradable polymers [27]; thus, *Rhodococcus* degrading potential has been assessed for all genes related to PBAT, Ecoflex, PHB, PHA, PLA, PCL, PPL, PBS, PBSA, PES, PHBV, PHPV, and PHV degradation. The considered input genes encode PBS depolymerase, PHB-depolymerase, cutinase, esterase, PET-hydrolase, PLA-depolymerase, lipase, MCL-PHA-Depolymerase, polyesterase, hydrolase, and carboxylesterase. Figure 12 shows the results of the bioinformatic analysis where the number of unique HIT sequences identified in *Rhodococcus* genomes is reported for different enzyme categories putatively related to polyester degradation. The three enzymatic classes with the highest number of unique HIT sequences of *Rhodococcus* species belong to carboxylesterase (7007 sequences), PLA-depolymerase (5302 sequences), and PHB-depolymerase (4328 sequences) groups.

In this case, a clusterization tree was also generated, showing that all *Rhodococcus* putative polyester-degrading sequences were clusterized into a single tree against the reference gene inputs (Appendix A). 

### 3.5. Dominant Rhodococcus Species and Main Enzyme Classes for Plastic Polymer Degradation

In order to predict the role in plastic degradation of different *Rhodococcus* species, the species presenting the most abundant number of unique sequences with a putative biodegradative potential (Figure 4) were used for score calculations. They were mostly isolated from diverse environmental niches, such as soils, contaminated environments, and water systems, some of them are pathogenic bacteria, and others are often detected in human infections. Since seventeen *Rhodococcus* species showed the highest number of sequences (although the different number of genomes for each *Rhodococcus* species), they were used for the following analyses, and they were: *R*. *hoagii*, *R*. *ruber*, *R*. *pyridinivorans*, *R*. *qingshengii*, *R*. *opacus*, *R*. *erythropolis*, *R*. *aetherivorans*, *R*. *triatomae*, *R*. *fascians*, *R*. *jostii*, *R*. *rhodochrous*, *R*. *koreensis*, *R*. *equi*, *R*. *corynebacterioides*, *R*. *kroppenstedtii*, *R*. *wratislaviensis*, and *R*. *globerulus*. The score was based on *Rhodococcus* sequence similarity with respect to the diverse gene targets chosen for the degradation of three groups of polymers (C−C backbone plastics, heteroatomic backbone plastics, and polyesters) and the sum of the scores resulted in Figure 13. It is noteworthy that among the 45 analysed *Rhodococcus* species, three out of four species often associated with pathogenic strains (*R*. *hoagii*, *R*. *equi*, and *R*. *fascians*) showed a high number of degrading genes and additionally high score levels. The highest scores are related to C−C backbone plastic-degrading sequences; in particular, *R*. *hoagii* showed the highest number of these enzymes despite a higher number of HIT sequences retrieved in *R*. *equi*.

Among human infectious-associated species, *R*. *triatomae* presented the highest scores, especially with respect to C−C backbone plastic gene products.

Considering the environmental-associated isolated species, they presented average similar score levels, with some exceptions: *R*. *ruber* seemed to present the highest score value for PET and PU gene products despite the low number of identified HIT sequences with respect to other environmental-associated species; *R*. *pyridinivorans* and *R*. *qingshengii* revealed the highest scores towards PE degrading genes, and *R*. *wratislaviensis* showed the lowest scores towards polyester degrading genes. In this respect, the lowest score levels were detected for all the species against polyesters which were the highest polymer category in terms of the number of identified sequences.

Analyzing the score of the different enzymes identified for the biodegradation of synthetic and biodegradable plastics, multicopper oxidases, alkane 1-monooxygenases, and cytochrome P450 hydroxylases were the enzymes of major interest for C−C backbone plastic group (Figure 14). Alkane 1-monooxygenase class possesses the highest score value for *R*. *erythropolis* and *R*. *qingshengii*; multicopper oxidase class is prominent in *R*. *erythropolis*, *R*. *qingshengii*, *R*. *opacus*, and *R*. *equi*; cytochrome P450 hydroxylase category presented the highest score values for the *R*. *opacus* and *R*. *erythropolis* species. Worthy of being specified is the apparent contradiction for *R*. *hoagii* score values for C−C backbone plastic-degrading genes. Probably, the considerable score levels are primarily due to other enzymes considered, such as transporters and membrane proteins.

The scores of the other two plastic-degrading enzyme categories related to heteroatomic backbone plastics and polyesters were simultaneously analyzed because of the overlapping capabilities of these enzymes (Figure 15). Above all, carboxylesterase showed the highest scores and, in particular, *R*. *erythropolis*, *R*. *qingshengii*, *R*. *opacus*, and *R*. *equi* were the most prominent species. A similar score-species pattern was observed for *para*-nitrobenzylesterase, PU esterases, and esterases, with the exclusion of *R*. *qingshengii*. PLA-depolymerase presented the highest score values for *R*. *fascians* species. The scores for PHB-depolymerase were mostly the highest in *R*. *erythropolis*, *R*. *qingshengii*, *R*. *opacus*, *R*. *rhodochrous*, and *R*. *pyridinivorans*. On the other hand, cutinases presented average low score values; the highest was shown for *R*. *erythropolis*, *R*. *qingshengii*, and *R*. *fascians* species. 

## 4. Discussion

Increasing plastic production and consumption raised the consequent issue of its disposal management, which is becoming a compelling forefront [23,45]. For this reason, biodegradation of synthetic plastic has been increasingly studied, including the feasibility of bacteria/enzymes biodegradative processes [21,26,27,46,47]. In this context, among bacterial genera with extraordinary biodegradative capabilities, *Rhodococcus* genus was selected for its remarkable metabolic capabilities and strong persistence under stress conditions [14]. For the first time, in this work, *Rhodococcus* genomes were analyzed to predict genetic determinants with degrading capabilities towards synthetic and biodegradable plastics categorized into three classes: (i) C−C backbone plastics, (ii) heteroatomic backbone plastics, and (iii) polyesters. Comparative bioinformatics analysis was based on the selected gene products from several microorganisms related to plastic degradation. Their comparison against 669 *Rhodococcus* genomes showed that around 57% of unique HIT sequences were covered by polyester-like degrading enzymes (mostly PBAT and PHB), while 24% and 18% included C−C backbone plastics (PE) and heteroatomic backbone plastics groups (PET and PU), respectively. Preliminary analyses revealed amino acid sequences possessing peptide signals for extracellular secretion. This suggests their putative key role in the plastic degradative processes [17]. Remarkably, these results reflect the *Rhodococcus* genus biodegradative potential towards biodegradable plastics, especially towards PBAT and PHB. These outcomes are also evidenced by analyzing the single *Rhodococcus* species with the exclusion of the ones with the lowest number of HIT sequences. 

On the other hand, score values highlighted PE-degrading sequences as the most probably involved especially for *R*. *pyridinivorans*, *R*. *qingshengii*, and *R*. *hoagii* genera [48,49], which have the highest PE-degrading potential. With respect to C−C backbone plastic gene products, *R*. *triatomae* presented the highest score if we consider only human-associated species, as mostly reported in the literature [50,51]. This high similarity for PE-degrading enzymes is probably due to the *Rhodococcus* strain origin of the majority of input sequences related to this class of polymers [17,31]. The association of the diverse species of *Rhodococcus* genus to diverse niches could be a useful hint to predict the most promising ecological niches, where rhodococci could be prominent in plastic degradation.

The subtree clustering analyses for PE-plastic gene products showed that the majority of homologous enzymes with respect to multicopper oxidases, alkane monooxygenases, or cytochrome P450 hydroxylases of diverse *Rhodococcus* species belong to the following species: *R*. *opacus*, *R*. *wratislaviensis, R*. *marinonascens*, *R*. *imtechensis*, *R*. *jostii*, *R*. *koreensis, R*. *rhodochrous*, and *R*. *aetherivorans*. In this regard, PE is considered one of the most used and wasted plastic due to its properties of non-degradability and durability [52], and only a few bacterial strains possess the ability to degrade, at least partially, diverse types of PE [17,29,30,31]. For this reason, these indications could be valuable for further investigations of the most interesting species; PE biodegradation could be performed by oxidation mediated by multicopper oxidases, probably through extracellular oxidation due to the preliminarily identified peptide signal sequences for extracellular secretion, and then smaller PE fragments can be internalized by transporters and other oxidative systems, including alkane monooxygenases or cytochrome P450 hydroxylases, could participate in further steps of intracellular PE-smaller fragment oxidation [18].

*Rhodococcus* degrading potential for plastics with heteroatoms in the main chain showed that the highest number of identified unique HIT sequences belong to PU esterase and *para*-nitrobenzylesterase and with a lower extent to cutinase. These enzymatic classes could putatively cover PET and PU biodegradation. Indeed, enzymatic hydrolysis is the route for the degradation of both these aromatic polyesters [21,34,53].

Although sequence clustering showed that PU degrading enzymes belonging to *R*. *opacus*, *R*. *wratislaviensis*, *R*. *jostii*, *R*. *aetherivorans,*
*R*. *hoagii, R. erythropolis*, *R*. *qingshengii*, *R*. *rhodochrous*, *R*. *zopfii*, *R*. *triatomae*, and *R*. *fascians* were the closest to the reference, *R*. *ruber* presented the highest score value for both PET and PU gene products (despite the low number of identified HIT sequences with respect to other environmental-associated species).

Some categories of polyesters are considered biodegradable [27], comprising PBAT and PHB, which represent the polymers associated with the highest number of unique HIT sequences. In literature, the enzymes reported for the ability to degrade polyester-based plastics include polyester depolymerases, esterases, carboxylesterase, cutinase-like enzymes, and lipases [26,27]. In line with these results, the three main enzymatic classes predicted for their involvement in polyester biodegradation of *Rhodococcus* species belong to carboxylesterase (42% of total polyester predicted degrading enzymes), PLA-depolymerase (32%), and PHB-depolymerase (26%). Consistently, carboxylesterase, PLA-depolymerase, and PHB-depolymerase showed the highest scores for certain species. On the other hand, cutinases are present in low abundance among unique HIT sequences of *Rhodococcus* genomes, showing an average low score value.

Overall sequence comparison and score calculation were useful tools to weigh different *Rhodococcus* species and different genes putatively involved in the metabolism of diverse plastic polymers. In addition, tree clusterization allowed us to comprehend and distinguish the putative role of the new genes identified for the diverse *Rhodococcus* species. Notably, the combination of the similarity scores of enzymes from diverse microorganisms against *Rhodococcus* genomes with the clustering analysis poses the basis for evolutional relationships studies, and in some cases, can suggest events of horizontal gene transfer of *Rhodococcus* identified sequences. 

The indications deriving from this study could be useful for future biotechnological applications at two levels: *Rhodococcus* strains can be exploited in plastic biodegradative processes and even some enzymatic functions could be further studied for catalytic reactions from a biotechnological point of view.

## 5. Conclusions

Since the current prominent plastic waste management problem at the global level, this research study aimed to shed light on *Rhodococcus* genus traits for its well-known remarkable biodegradative capacities. This work contributes to gain novel knowledge on the biodegradative potential of *Rhodococcus* species towards both synthetic and biodegradable polymers mediated by specific enzymatic classes. New insights on primary enzymatic classes involved in most produced and utilized plastic materials were shown. The sequence predictions derived from this study can be promising as an explorative survey to base the further sequence utilization as a reference for plastic contamination biomonitoring and the consequent plastic removal with biotechnological applications. 

## Figures and Tables

**Figure 1 microorganisms-10-01846-f001:**
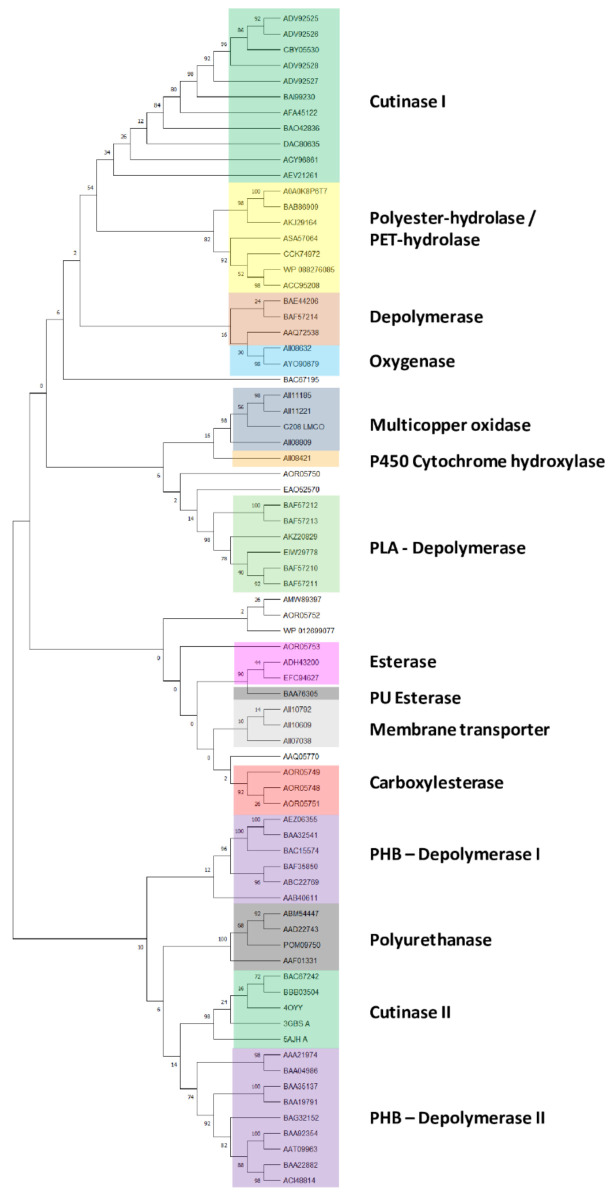
Clusterization tree of unique input gene products belonging to different microorganisms (retrieved from PlasticDB database, literature data, *omic* studies, and functional analyses for the capability of degrading at least one type of plastic) divided into fifteen clades: cutinase I (clade I), polyester-hydrolase/PET-hydrolase (clade II), depolymerase (clade III), oxygenase (clade IV), multicopper oxidase (clade V), cytochrome P450 hydroxylase (clade VI), PLA-depolymerase (clade VII), esterase (clade VIII), PU esterase (clade IX), membrane transporters (clade X), carboxylesterase (clade XI), PHB-depolymerase I (clade XII), polyurethanase (clade XIII), cutinase II (clade XIV), and PHB-depolymerase II (clade XV).

**Figure 2 microorganisms-10-01846-f002:**
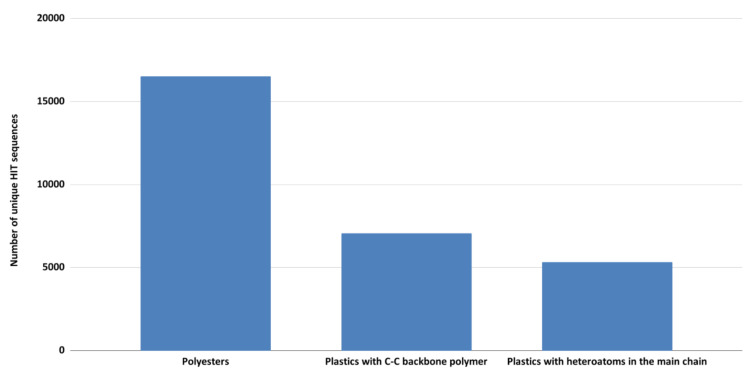
Total number of unique HIT sequences, deriving from the comparison of 124-input degrading-genes against *Rhodococcus* genomes, were classified into three main polymer categories: polyesters, C−C backbone plastics, and heteroatomic backbone plastics.

**Figure 3 microorganisms-10-01846-f003:**
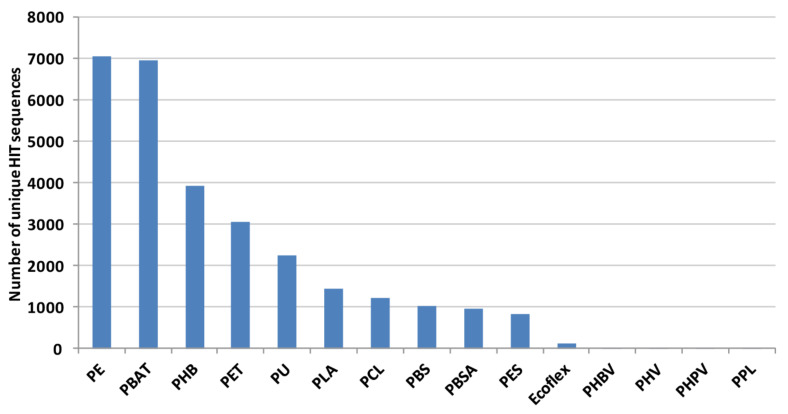
Total number of unique HIT sequences identified in *Rhodococcus* genomes for each polymer material.

**Figure 4 microorganisms-10-01846-f004:**
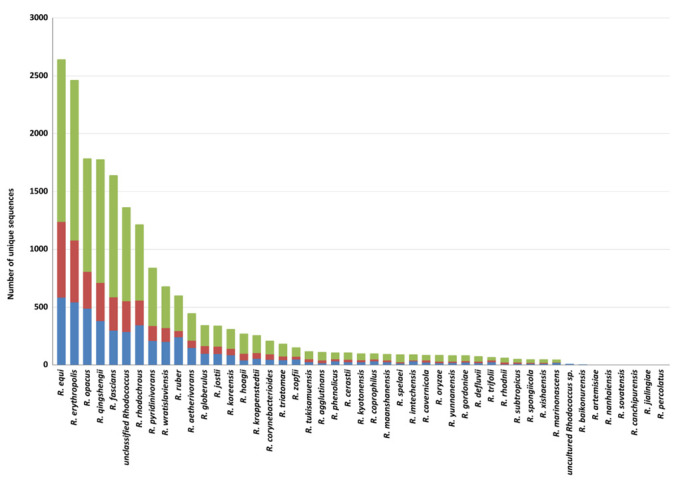
Total number of unique HIT sequences identified in *Rhodococcus* species genomes for each polymer category: polyesters, heteroatomic backbone plastics, and C−C backbone plastics.

**Figure 5 microorganisms-10-01846-f005:**
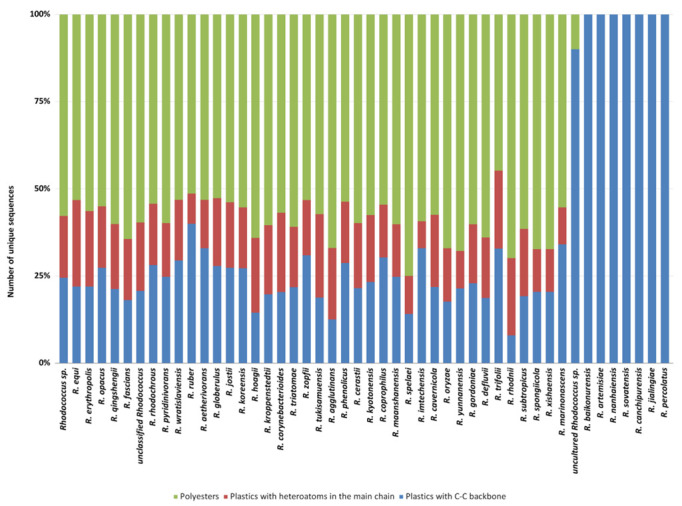
Normalized number of unique HIT sequences identified in *Rhodococcus* species genomes for each polymer category: polyesters, heteroatomic backbone plastics, and C−C backbone plastics.

**Figure 6 microorganisms-10-01846-f006:**
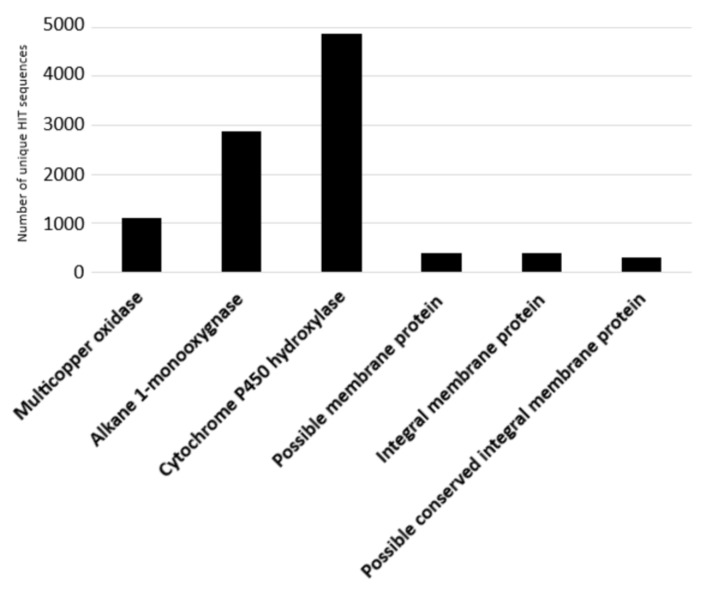
Number of unique HIT sequences identified in *Rhodococcus* genomes for different enzyme categories putatively related to C−C backbone plastic degradation.

**Figure 7 microorganisms-10-01846-f007:**
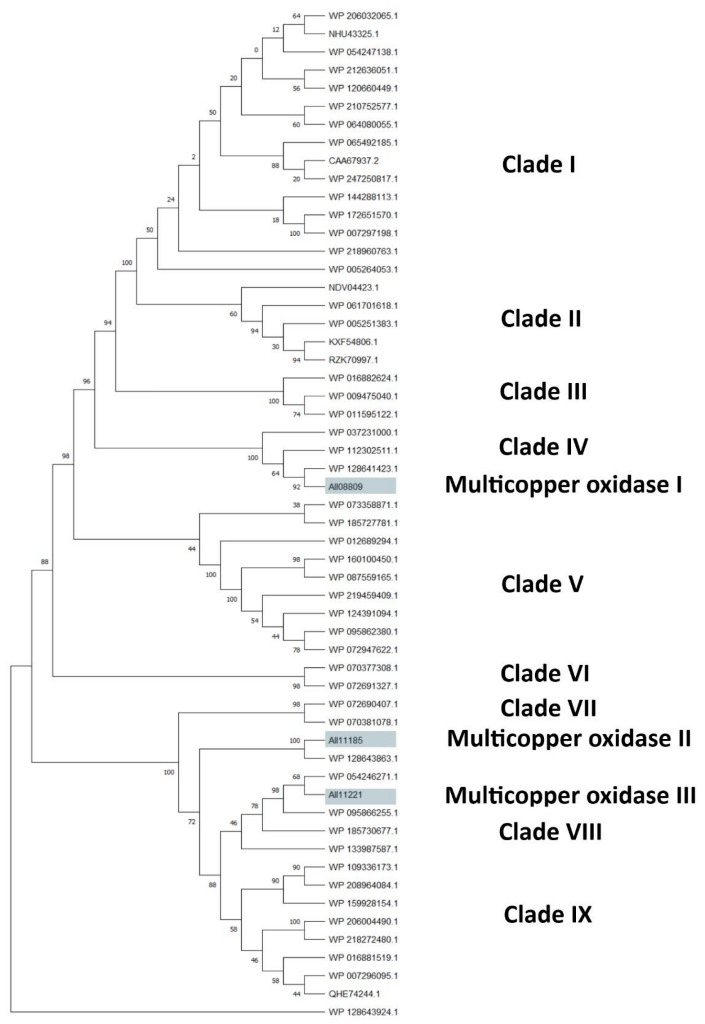
Subtree clusterization of HIT sequences identified by oxidase input genes putatively involved in C-C backbone plastic degradation. Marked boxes evidence the reference input genes.

**Figure 8 microorganisms-10-01846-f008:**
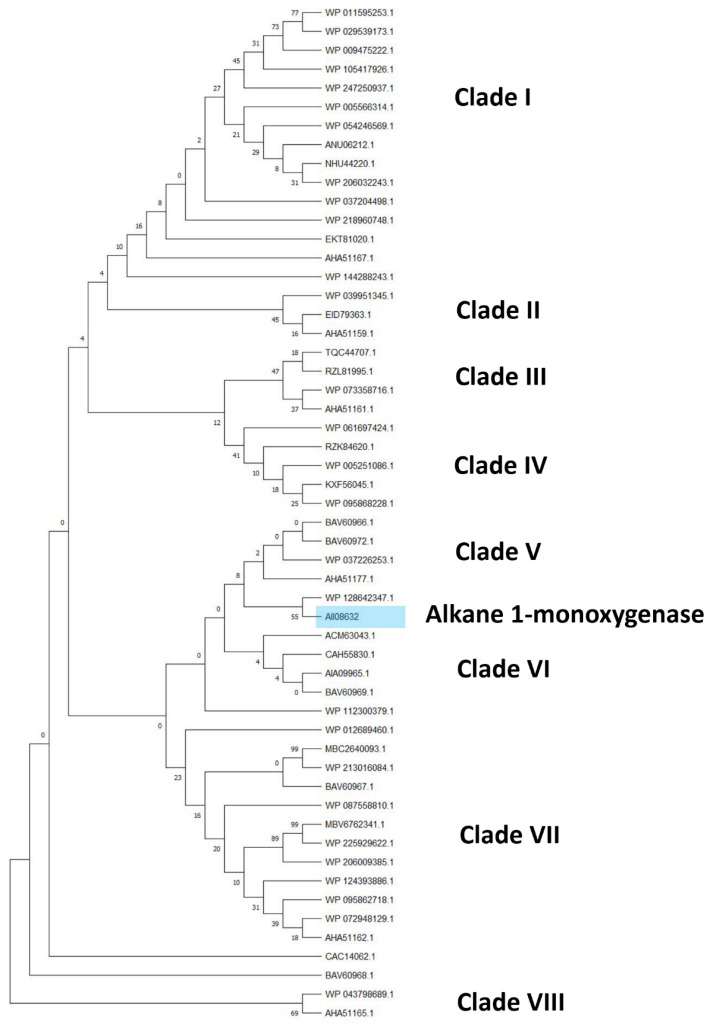
Subtree clusterization of HIT sequences identified by alkane monooxygenase input genes putatively involved in C-C backbone plastic degradation. Marked box evidences the reference input gene.

**Figure 9 microorganisms-10-01846-f009:**
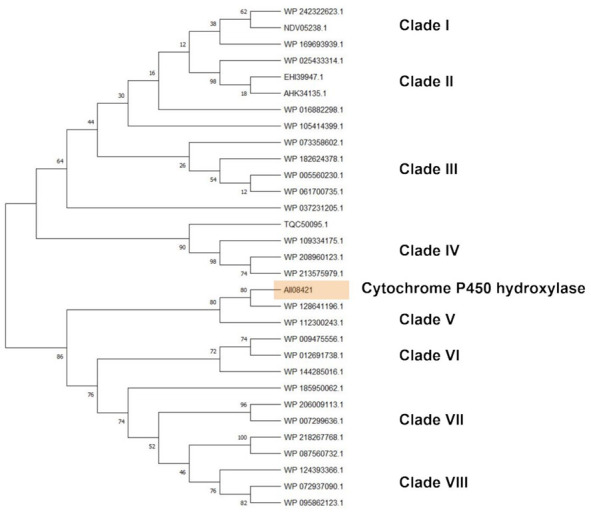
Subtree clusterization of HIT sequences identified by cytochrome P450 hydroxylase input gene putatively involved in C-C backbone plastic degradation. Marked box evidences the reference input gene.

**Figure 10 microorganisms-10-01846-f010:**
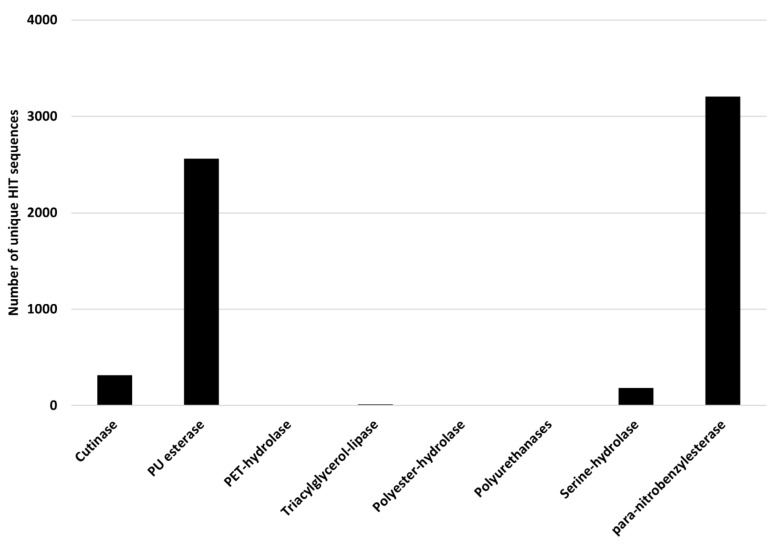
Number of unique HIT sequences identified in *Rhodococcus* genomes for different enzyme categories putatively related to heteroatomic backbone plastic degradation.

**Figure 11 microorganisms-10-01846-f011:**
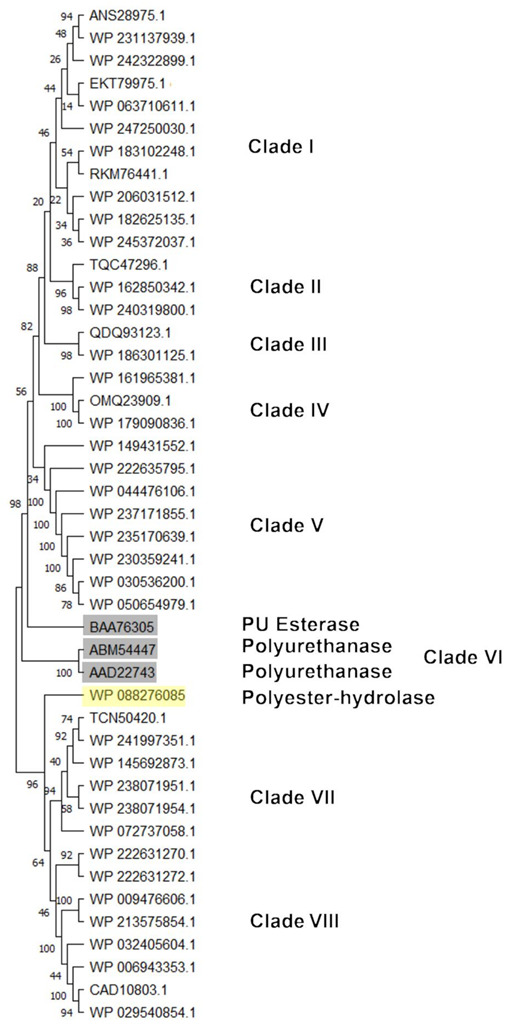
Subtree clusterization of HIT sequences identified by PU esterase and polyurethanase input gene putatively involved in heteroatomic backbone plastic degradation. Marked boxes evidence the reference input genes.

**Figure 12 microorganisms-10-01846-f012:**
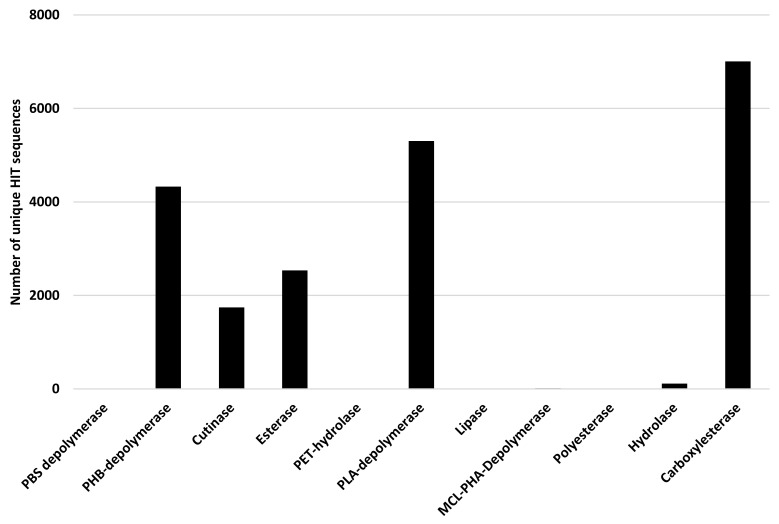
Number of unique HIT sequences identified in *Rhodococcus* genomes for different enzyme categories putatively related to polyester degradation.

**Figure 13 microorganisms-10-01846-f013:**
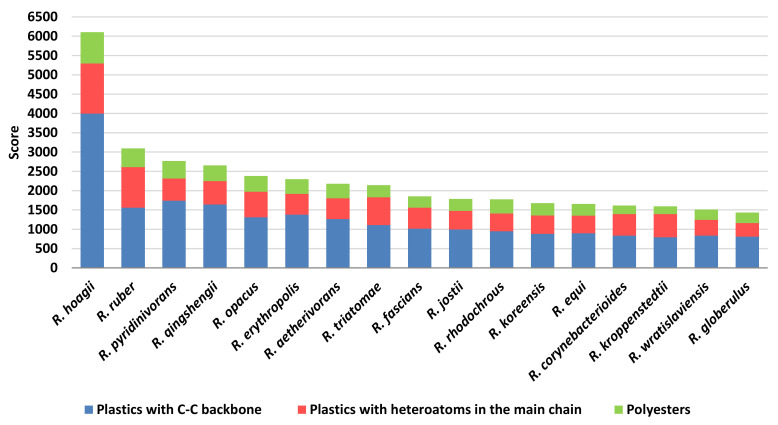
Score values for the most prominent *Rhodococcus* species for each class of diverse polymer materials.

**Figure 14 microorganisms-10-01846-f014:**
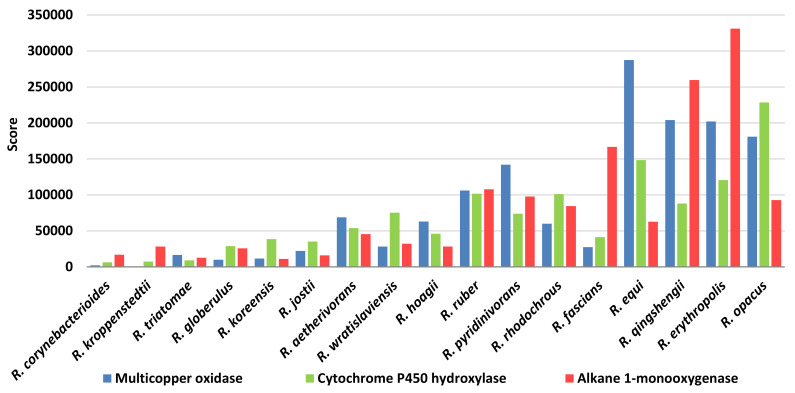
Score values for the most prominent *Rhodococcus* species for the main enzymatic categories involved in C-C backbone plastic degradation.

**Figure 15 microorganisms-10-01846-f015:**
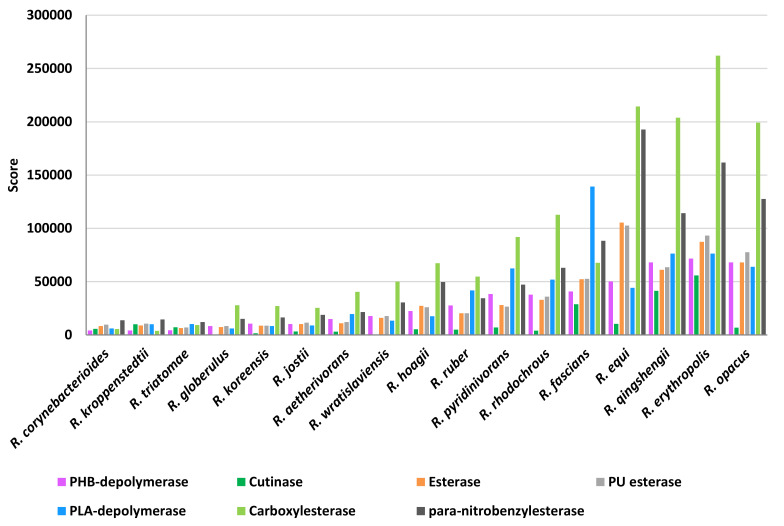
Score values for the most prominent *Rhodococcus* species for the main enzymatic categories involved in PET, PU, and polyesters degradation.

## Data Availability

All data generated during this study are included in this article.

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
