# Peer review of "Genome-Based Exploration of *Rhodococcus* Species for Plastic-Degrading Genetic Determinants Using Bioinformatic Analysis"

_microorganisms, 2022, doi:10.3390/microorganisms10091846_

Round 1

Reviewer 1 Report

The authors focused on Rhodococcus species, which are capable of degrading various organic and persistent compounds, and estimated the genes involved in the biodegradation of synthetic plastics through bioinformatics analysis using information from databases, literature data, and ohmic approaches. I think this is an excellent paper that focuses on the plastic degradation of Rhodococcus species, which have not been analyzed in detail before, and estimates their degradation ability. I understand that this is an important study with a very future advantage. In my opinion, it requires minor revision before ready for publication. I hope you will consider the following points of concern. 

I will appreciate it if you accept suggested corrections and resubmit your revised manuscript.

Major comments

Are enzymes you estimated secreted out of the cell for degradation of the plastics? The findings on this have not been described. The authors would need to consider this. In addition, the authors would mention whether the enzyme has a signal sequence for extracellular secretion.

If possible, please mention in what environment the presumed gene is transcribed. Then, the authors should mention whether their transcription is inductive.

Reviewer 2 Report

The paper is well written and adds to existing literature on the subject. The introduction is well composed and has been developed on the right lines. Most appropriate methodology has been used for analysis of data and information. The results have been reported well. Logical sequence of interpretation has been followed and developed scientifically. The discussion has been well brought out and the cogency of arguments are well thought out. The discussion is comprehensive and complete. References are as required. May be accepted for publication.

Reviewer 3 Report

In the work, Rhodococcus genome mining for the prediction of plastic-degrading enzymes was conducted and the high plastic-degrading potential of the certain species was shown. Have you analyzed the literature data on the available practical studies of plastic degradation by the rhodococci strains and how does this correlate with your data on the genomes analysis of individual species?

Minor comments

In the introduction, please highlight more specifically the purpose of the work.

Please improve the resolution for Figures 1-9, 11.

For Figure 11: On the subtree, highlight the references sequences.

Line 193 The paragraph begins by mentioning 16 polymer materials, but lists three groups with examples, probably just 16 examples? Then it is necessary to list them without dividing into groups, which has already been done above, or not to mention the exact number. Please rephrase the paragraph.

Line 250 Typo “metablisms“ should be ”metabolisms"

Line 323-328 Reference proteins are not named in the text, only their GenbankID.

The references part must be written according to the requirements of the journal, where the numbering without square brackets.

Reviewer 4 Report

L1-2: Title should be reconsidered to fit the research contents well. At least, "new insights" is not fit the contents, rather confusing.

Introduction section: Lengthy and circumlocutory overall, obscure essential points. Too many paragraphs, like a review article. Revise to make Introduction compact (minimum) by deleting unnecessary sentences and/or paragraphs. For example, L100-102 can be included in the first paragraph.

L45: Delete “at present”.

L69: Grammatical error “This category include”

L116 and other parts: Specify “around 670”.

L126-127 and other parts: PHA is a genetic term for PHB, PHV, etc. Parallel use of PHA and individual polymers within PHA makes confusing and is inappropriate.

L145-152: Too many small paragraphs should be combined.

L175: Delete “A number of”.

L184-185: Revise “P450 cytochrome” to “cytochrome P450”.

*Fig. 1 and other parts: As an important point (prerequisite) to verify the usefulness of this study, the enzymes within the listed categories can invariably degrade plastics? The reviewer cannot find the validity of this prerequisite.

L195-198: Definition of abbreviations in main text should be done only once. Abbreviations, PHB, PHV, PHBV, PHPV, are already used upper parts.

*L225-230: The comparison here among Rhodococcus species is confusing. The results depend on the genome numbers used for each species. The comparison based on the sum is not appropriate. For example, the relative number in a genome should be used for comparison. In addition, information about the used genomes is lacking. Are all genomes complete ones? If incomplete genomes are included, the results must be carefully interpreted.

L260-262: Paenibacillus and Rhodococcus are phylogenetically distant at the phylum level. Thus, the description is lack of validity, and should be deleted.

Figs. 1, 7-9 and 11: Insufficient explanations of phylogenetic trees.

L304-310: Both the fusion between heteroatoms and degradation of a molecule containing heteroatoms are included; and thus the contents are confusing. Careful explanations are necessary.

L361: Why 45 species? In above paragraph, only 17 species are considered.

L409-422: The first paragraphs in Discussion are overlapped with the contents in Introduction. Avoid repeat.

L423-437: The third paragraph is the repeat of methods and results. Remove or minimize it.

L441-442 “These results…Rhodococcus genoms”: As commented elsewhere, this is not supported by the results in this study.

L446-447 and relevant parts: How can they be divided clearly? Pathogenic species should also be found in the environment. Division of pathogenic and environmental species is obscure and problematic. In addition, why should pathogenic and environmental species be divided? Is it important/necessary in the context of this study. The reviewer recommends to remove the contents relevant to pathogenic/environmental species. Or, at least the correlation between the presence of pathogenic genes and the target genes in this study must be shown.

Round 2

Reviewer 4 Report

Corrections in English writing are required especially in the parts that made revisions.